# How Effective Is a Buddhist Pilgrimage Circuit as a Product and Strategy for Heritage Tourism in India?

**Kiran A. Shinde**

Department of Social Inquiry, School of Humanities and Social Sciences, La Trobe University, VIC-3086, Australia; k.shinde@latrobe.edu.au

**Abstract:** Buddhist heritage in India is receiving reasonable attention for tourism development with a particular emphasis on promoting Buddhist circuits. One Buddhist pilgrimage circuit covering eight locations including four most sacred places, namely, Bodhgaya, Sarnath, Kushinagar, and Lumbini, is popular for spiritual and religious reasons. Fieldwork conducted in these four sites reveals complex patterns of visitation that question the idea of a circuit as a tourism product. The primary circuit-goers are foreign Buddhist followers, but the magnitude of their visitation is very low. They constitute less than 10 percent of visitors and are far outnumbered by domestic visitors. Domestic visitors driven by recreational purposes hardly complete the entire circuit; their visits are directed to two popular sites while other sites are used as "middle-of-the-trip centers" or places for daytrips. The interviews with various stakeholders including religious institutions, private tour operators, and relevant government agencies, show that the itineraries of circuit-goers depend on several factors including the convenience of travel, accommodation facilities, the ability to perform pilgrimage rituals, and institutional governance for tourism. This paper argues that Buddhist pilgrimage circuits operate more as a cultural landscape at a cognitive level rather than as a distinct physical route and a product that can be effectively translated into as strategy for tourism development in Buddhist sites.

**Keywords:** pilgrimage circuit; buddhist heritage; buddhism; tourism; bodhgaya; India

## 1. Introduction

Buddhist tourism across Asia has gained considerable attention in recent scholarship. Several studies dwell on how Buddhism provides a variety of resources for tourism including monastic heritage, Buddha's teachings, Buddhist philosophies, and places where seekers can fulfil spiritual quests [1]. Many of these studies report that the traditional practice of pilgrimage has continued in many Asian countries where Buddhism is popular and one can find more active practice of religion [2,3]. However, these are places where Buddhism reached much later in the historical spread of the religion and their pilgrimage traditions are different from those that are found in India which is regarded as "the heartland of Buddhism" [4,5].

Buddhism, founded by Buddha around the sixth century B.C.E in the Indian subcontinent, went through cycles of glory and decline and thus exhibits a complex Buddhist pilgrimage landscape that is different from other Asian countries. Buddhism travelled to other parts of the Asian subcomponent and assimilated with the local faiths and indigenous practices. However, in India, it was lost for many centuries generating a landscape that was more of archeological and historical in nature rather than a place of active religious practice. In the process, several places related to Buddha evolved into major pilgrimage centres for Buddhist followers. India has probably the largest concentration of archeologically and historically important Buddhist sites. But ironically, Buddhists account for only 0.8% of the 1.35 billion people in India and this makes up for the uniqueness of the situation of Buddhist tourism [6]. It is largely the international Buddhist followers that have led to the recovery of Buddhist sites and Buddhist heritage for tourism.

With the resurging interest in Buddhism globally, sites and trails related to Buddha in India are being rediscovered and packaged for Buddhist tourism [7–9]. In many iconic sites, there has been considerable increase in the numbers of international Buddhist followers. This increase, many believe is also because of proliferation of Buddhist monasteries that are being built by international Buddhist societies and associations to continue the practice of Buddhist traditions [9]. Riding on this growth, many stakeholders are participating in promoting Buddhist circuits: religious institutions (monks and monasteries) are reviving and nurturing traditional pilgrimage trails; private tour operators are offering package tours; and national and state governments are emphasizing development of pilgrimage circuits in tourism policies and projects.

This paper has two research questions: (a) do traditional Buddhist pilgrimage circuits make for a good religious tourism product, and (b) whether pilgrimage circuits are an effective strategy to promote Buddhist heritage for tourism in India. To find answers to these questions, the paper first examines the articulation of the main Buddhist pilgrimage circuit in terms of places, place-stories, practices, and itineraries based on religious traditions. It uses the conceptual approach of cultural landscape and the geographical underpinnings of a circuit to explain the articulation of Buddhist pilgrimage circuits. Then, it investigates the visitation patterns in the different sites that constitute the pilgrimage circuit to understand if the pilgrimage circuit really works as a circuit. While doing so, the paper also challenges the assumptions about incorporating the idea of the traditional pilgrimage circuit as a main strategy into the tourism policy. By answering these questions, the paper offers a nuanced understanding of Buddhist pilgrimage circuits and their relevance for tourism in India–both as a concept, and as a practical strategy, for tourism development in Buddhist sites.

This paper is based on the findings from the fieldwork that was conducted in December 2019 in four key sites of Buddhist pilgrimage circuit namely Bodhgaya, Sarnath, Kushinagar, and Lumbini (in Nepal). Fieldwork included participant observation, visitor surveys, review of government reports and interviews with key stakeholders. A total of 75 interviews were conducted with government officials, monks, managers of monasteries, hotel owners and managers, tour operators, shopkeepers, and local community leaders.

The remainder of the paper is organised in seven sections. Following this introduction is a brief exploration of key concepts of pilgrimage circuits as cultural landscapes. The third section provides an overview of history of Buddhism that has led to creation of a palimpsest of Buddhist heritage including sites and circuits for pilgrimages. Pilgrimage-circuits is the focus of the next section where circuits framed by the state, examples of those offered by private tour operators and instances of traditional pilgrimages organised by religious institutions are discussed in great details. These circuits are then validated with field-based data on visitation patterns in Buddhist sites and insights from interviews with key stakeholders. The final section offers concluding remarks and recommendations for sustainable promotion and development of Buddhist tourism in India.

## 2. Understanding Pilgrimage Circuits

The idea of a Buddhist circuit frames a certain religious and cultural imagination of the land of Buddha [10]. Such articulation is germane to most organised religions where places are brought into an assemblage where the natural and geographical features are associated with the legends of their founders; for example, the Holy Land of Jesus for Christians, the Mecca-Medina complex of the Prophet for Muslims, and so on. Many religions and faiths promote their sacred places using place-narratives related to their prominent deities, gods, goddesses, and other divine objects. Such pilgrimage sites are often explained using the idea of cultural landscapes.

In simple terms, cultural landscapes result from the action of human beings on nature and natural forms and represents a plethora of relationship between humans and their environments. However, in case of pilgrimage sites such action also has metaphysical elements of sanctity and divinity that are imbued in the landscape through conscious

efforts [11,12]. Several studies of pilgrim-towns and pilgrimage trails suggest that the following concepts are key to explaining a pilgrimage landscape, and by extension to pilgrimage circuits.

*Sacred geography*: Sacred geography refers to the ensemble of physical and geographical features of a pilgrimage place. These include natural elements such as river, trees, and ponds, sanctified artefacts and objects, and temples; each is identified with a physical representation of a deity and has myths and legends that contextualise them as sacred [12–14]. A religious framework is developed that tie these elements with rituals and performances for devotees of the deity/deities. A boundary around this ensemble or sacred territory is also considered sacred and its circumambulation becomes an integral ritual of reinforcing the sacred character of the place.

*Place-narratives*: A place is remembered through the stories that are attached to it. The place does not mean a setting alone, in fact it constitutes the narrative and memory. Spatial stories that are intimately connected with physical places are very important as it is when they are talked about, recounted, or written and depicted in a certain way, they lead to a better appreciation of the landscape [15]. They take many forms from origin-stories to location-incident based miracle-stories. For instance, in the pilgrimage landscape of Braj which is "conceived of as a forest" [14], pilgrims walk to hear stories of Krishna's miracles in forests which reinforces and perpetuates their belief in Krishna [16].

*Symbols*: The landscape can be appreciated at both the physical and metaphysical level for its religious symbolism and meaning. Symbols in cultural landscape have the power to compress complicated meanings into a specific object or behaviour. They act like an environmental and artefactual archive that can be decoded by physical contact [15,17]. Haberman cites of several examples in Braj where pilgrims connect with symbolic elements such as the rock at Charan Pahari which symbolises footsteps of Krishna. [14].

*Journey, route, and timing*: The idea of journeying-taking time to move and following a route is essential part of knowing a cultural landscape. Kinsley notes that by travelling in a sacred landscape, pilgrims "open themselves to the sacred power, the numinous quality, of the landscape, whereby they establish a rapport with the land that is spiritually empowering [17]." In case of Braj, a route that takes a pilgrim on a journey of about 23 to 40 days through "forests, lakes, ponds, and mountains will lead to an experience of Krishna's presence in the landscape" [16]. According to Haberman, "the physical geography of Braj is itself a kind of text, and ... the preeminent way of 'reading' this text is by means of pilgrimage" [14].

Do these elements of sacred geography, place-narratives, and symbols translate into a tourism circuit? For that, it is first necessary to see how a circuit is identified for tourism.

At the concept level, circuits provide a mechanism for doing tourism and represent "temporal spatial-carriers of the tourist experience" [18]. In the itineraries of a circuit all components of tourism such as lodging, boarding, travel, attractions, sites are interconnected systematically and as such these itineraries can be sold as tourism products. Mostly such itineraries are imagined and have a "nonmaterial form" [9]. However, realising the itinerary is essentially a physical act. Buddhist circuit in India that has become popular in tourism itineraries is based on "a set of known, fixed sites associated with a singular tradition of Buddhism" that resulted from "archaeological rediscovery and restoration" [9]. Geary further proposes that circuits are "containers of current and flow" and represent "key conduits for interconnection where the movement of people can be managed, serviced and coordinated through a concentration of sites and infrastructure" [9].

From the above theoretical standpoints, one can speculate that the idea of cultural landscape (from a cultural and geographical perspective) may be translated into a tangible tourism product (an economic activity focused on efficiency and experience) as is being done in case of Buddhist circuits. However, there will be challenges as will be argued in this paper as it explores Buddhist pilgrimage circuits which is gaining traction as a major strategy for promoting and managing Buddhist tourism in India.

### 3. Buddhism and Buddhist Pilgrimage in India

Buddhism prevailed as a major religion in India for many centuries beginning from the lifetime of Buddha itself (3rd BCE). The religion spread across the sub-continent under the leadership of the Mauryan King Ashoka who erected thousands of Buddha shrines, patronized Buddhist monks to start monasteries and centers of learning, and sponsored pilgrimages to other kingdoms. Subsequent rulers continued this patronage and many Buddhist monks including *Śāntarakṣita, Padmasambhava*, and *Atisa* took Buddhism to other parts of the subcontinent. Over time Buddhism splintered into a few sectarian traditions and schools but it also was assimilated into the already existing indigenous religious practices. Thus, a rich syncretic heritage related to Buddhism developed in amongst which pilgrimage travel by monks seeking Dharma and knowledge was considered meritorious [19].

In India, the organized religion of Buddhism waned around the thirteenth century due to loss of patronage and the invasion of the Turks who systematically destroyed monasteries [6]. As a result, Buddhism almost disappeared from most settlements in the plains pushing Buddhist followers to mountainous regions of Himalayas (which constitutes the present-day north-eastern region in India, and neighboring Bhutan and Nepal). During the early twentieth century colonial period, a wave of revival of Buddhism emerged due to a combination of interest of colonial powers in archaeology and antiquities and philosophical interest in the teaching of Buddha [10].

The anchoring point of this revival was the Maha Bodhi Society (Society of Great Enlightenment) that was founded by a Ceylonese monk (Anagarika Dharmapala) in 1891 to generate an international interest in Buddhism within and outside India. The society worked towards gaining control of the Buddhist shrine at Bodhgaya and this motivated the popularizing of Buddhist philosophy and religion as a glorious heritage of the country [6]. Another major contribution to resurgence of interest in Buddhism was the influx of Tibetan refugees under the leadership of the Dalai Lama in 1959. The exiled Lama established a government headquarters at Dharamsala in Himachal Pradesh and from there the Tibetan followers began to play an important role in reviving and internationalizing of Tibetan Buddhism across India and its neighbors [10].

As a tradition, Buddhist monks believe in following footsteps of Buddha and visiting as many places as possible related to Buddha in their journey towards spiritual enlightenment. As such, several places related to Buddha became established as pilgrimage sites shortly after his death [20]. At the core are four places mentioned in the authoritative text of *Maha Parinibbana Sutta*, where Buddha himself declares pilgrimage to four places as meritorious: Lumbini where he was born; Bodhgaya where he attained enlightenment under the Bodhi Tree; Sarnath where he delivered his first sermon, and Kushinagar where he died (the first is in Nepal while rest are in India). The other four also have place-stories related to Buddha: Rajgir is associated with several episodes of Buddha's life: his proselytizing of Emperor Bimbisar at Griddhakoota (Hill of Vultures); stay at Jivekarmavan monastery; and writing of His teachings at the first Buddhist Council. At Vaishali, Buddha stayed for long periods, preached his last sermon, and announced his Nirvana. Buddha spent a major part of his monastic life at Sravasti delivering the largest numbers of discourses and instructions including a popular miracle story where he presents himself as the million-fold manifestation seated on a thousand-petalled lotus with fire and water emanating from his body. Sankassa is a place where Buddha is believed to have returned to human realm after giving sermons to his mother in heaven. It is argued that it was Cunningham, first Director-General of Archaeology of India, who identified many Buddhist sites as forming a Buddhist circuit based on the translation of the text of Mahavamsa in1837 and his interaction with several Sinhalese Buddhist monks [20].

Over time, the popular conception of the Buddhist circuit includes "eight places of the Buddha": Lumbini, Bodhgaya, Sarnath, Kushinagar, Sravasti, Rajgir, Sankassa, and Vaishali as they have "become increasingly circulated and invoked in a variety of contexts" [9]. These places are widely seen as framing India's Buddhist landscape as well as locus of ritual practice for followers of Buddhism [10]. In Buddhism, the notion of a journey is

intrinsic to seeking and as such has evolved into an essential pilgrimage ritual for monks and laity. This remains the most authentic form of pilgrimage in Buddhist philosophy and is largely adopted by international followers and visitors which has led to a considerable emphasis on developing Buddhist circuit tourism. How Buddhist circuits are articulated by different stakeholders is discussed in the following section.

## 4. Methodology

The fieldwork was conducted in the main sites of the Buddhist circuit–Bodhgaya, Lumbini, Sarnath, and Kushinagar in December 2019. The fieldwork included participant observation and interviews with key stakeholders. The goal of the interviews was to understand the working of the pilgrimage circuit and how it intersected with the tourism in its constituent sites. As such the questions were asked around different elements of Buddhist heritage and Buddhist pilgrimage in those sites, patterns of visitation, rituals and performances undertaken by pilgrims, activities of tourists, roles of different stakeholders in the pilgrimage economy, the impacts of tourists on the local community, and the changes observed in Buddhist pilgrimages and pilgrimage sites over time. Using a snowball method (the author contacted the managers of temple/monastery trusts in these places and with their reference recruited potential interviewees), a total of 75 interviews were conducted covering representatives of the main stakeholders including 26 monks, 21 hotel managers, 12 government officials, 10 local community members, and 6 tour organizers. Interviewees were given a set of questions (between 13–18, with some customization for stakeholders) for interview. The 45 min interviews were conducted in person at the place of work of the stakeholders. The findings from the fieldwork are presented in the next section.

## 5. Existing Buddhist Pilgrimage Circuit/s

Based on the findings from the fieldwork, at least three stakeholders are identified presenting and promoting the Buddhist pilgrimage circuit: religious actors (monks and monasteries), private tour operators, and government agencies.

### 5.1. Religious Actors

The way Buddhist monks present the circuit, is best illustrated by a pilgrim-guide titled "Footprints of the Buddha (Pilgrimage to Buddhist India)" which was created by a monk named Ven. Bhikku Seelananda in 2010. The foreword clearly mentions that this is "a booklet for the pilgrims to be used as a handbook during the tour" (from February 10–23) but will also serve as a reference for future pilgrims because "a pilgrim with prior knowledge of the places and times in the life of the Buddha will develop a sense of awe and veneration upon visiting the sacred sites". The guide covers "only the most significant sacred places ... due to time and space limitations" and offers detailed descriptions of all the places for their association with Buddha and Buddhist discourses, their histories, key features, and a section on "things to be seen and venerated" at each place. According to this itinerary, the unidirectional tour starts at Sarnath which is considered as the birthplace of the sublime dhamma and follows through to Bodhgaya; Gijjakuta (Vultures' Peak); Rājagaha (Rajgir); Nalanda; Vaishāli (Vishālā); Kusinārā (Kushinagar); Lumbini, Kapilavastu, (in Nepal); Sāvatthi (Jetavanārāma) (or Sravasti); and ends in Sankassa and then pilgrims go out of the country from Delhi.

Although this pilgrimage circuit covers the eight places in one journey but it is not complete in the geographic sense of a closed circle-something which is a hallmark of religious circumambulations [9]. The circumambulation denotes covering of a sacred territory by following certain ritual protocols as found in many Hindu pilgrimages. However, starting and finishing at the same point is not necessary in Buddhist pilgrimages. Buddhist monks and monasteries have played a significant role in popularizing this route as they facilitate travel of their followers by providing lodging and boarding, spiritual counsel, and place for meditation and other Buddhist practices [21].

Other more recent religious institutions founded for promoting Buddhist teachings and philosophies also actively promote travel to abovementioned places but not necessarily in a circuit format. For instance, the Root Institute for Wisdom Culture which is located near Bodhgaya offers spiritual programmes including "several two or three-day pilgrimages throughout the season" that are "designed to facilitate an inner spiritual journey." The promotional text inviting potential pilgrims reads: "With our wealth of experience running pilgrimages radiating out from Bodhgaya, Root Institute can do all the groundwork, and offer accommodation&healthy vegetarian meals at Root Institute, take-away breakfast and/or lunch for all-day trips, arrange transportation, advice on relevant prayers&practices at each of the holy sites, and provide experienced pilgrimage guide (https://www.rootinstitute.ngo/spiritual-programme/pilgrimage accessed on 5 August 2020)."

In similar vein, many foreign monks who are ordained into Buddhism organise travel along the pilgrimage circuit to propagate teachings of Buddha (see Figure 1 that indicate two of such offerings). For instance, Ven. Robina, a monk based in America, believes that "going to these holy places touches people deeply, transforms them" as such pilgrimage is "authentic and led by practices recommended by Lama Zopa Rinpoche" (www.robinacourtin.com accessed on 5 August 2020). Similar pilgrimage packages are offered by many foreign disciples of Buddhism (see for example https://pariyatti.org/Pilgrimage/; www.buddhapath.com accessed on 5 August 2020). For instance, Pariyatti, a non-profit organization founded by Vipassana (a form of meditation) followers offers pilgrimage to smaller groups (generally about 30) of committed practioners of Vipassana and one of their key activities is that the participants "listen to stories and learn about the Buddha's life and teaching in the places where they actually happened . . . [and they] visit these sites not as sight-seers, but rather as site-sitters, [where they ] . . . meditate . . . to connect with the vibrancy of these locations as well as inspire and deepen [their] practice of vipassana" (https://pariyatti.org/Pilgrimage/Along-the-Path-India-Nepal-Pilgrimage accessed on 5 August 2020). In general, Buddhist circuit packages are mainly undertaken by foreign tourists interested in spiritual growth, wanting to experience meditation and yoga, and a general overall well-being based on Buddhist philosophy and teachings [21,22].

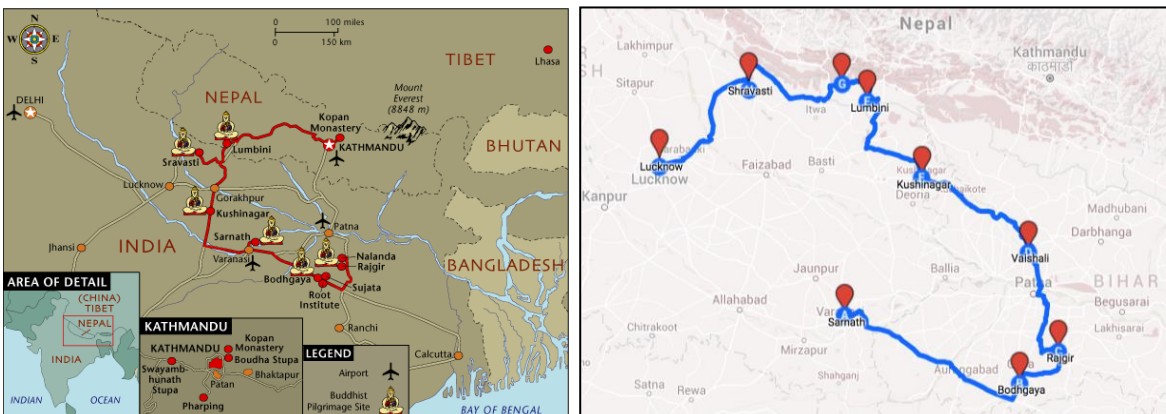

**Figure 1.** Buddhist pilgrimages offered by Buddhist teachers. Sources: www.robinacourtin.com, https://pariyatti.org/Pilgrimage/Along-the-Path-India-Nepal-Pilgrimage, accessed on 5 August 2020.

*5.2. Private Tour Operators*

The popularity of the circuit amongst foreign visitors motivated many commercial operators to offer travel packages around the circuit and these offerings have grown considerably in recent years. The websites of many operators show variations of routes and stay as the eight sites are packaged in different types of itineraries generating many sub-circuits depending on who is putting together the itinerary and who the target is. As shown in Figure 2, the inclusion of sites in the itineraries depend on the start and end of the

tour. A tour operator interviewed in Bodhgaya added how Delhi based itineraries (about 10 days) would typically include the iconic Taj Mahal in Agra in the package to make it attractive whereas in Kolkata based itineraries (typically 9 or10 days) a large amount of the time is spent on road travelling. In most cases the packages are predetermined with typical combinations of hotel stays and sightseeing.

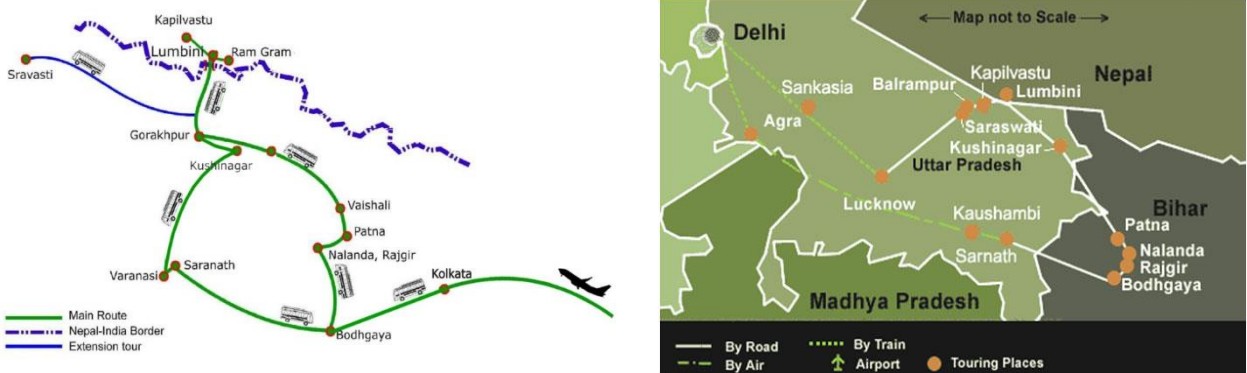

**Figure 2.** Two types of circuits offered by private tour operators. Source: http://www.buddhistcircuits.com/kolkata/, https://www.thetravelcare.net/mapof_cities.html, accessed on 5 August 2020.

*5.3. Government Agencies*

Government tourism departments have also realized the importance of promoting Buddhist circuits for tourism. The influx of international visitors promoted the government to introduce a "comprehensive, convenient, safe & reliable tour package in Buddhist circuit" with the launch of a special Air-Conditioned train called the "Mahaparinirvan Express"-special luxury train owned and operated by Indian Railways Catering & Tourism Corporation (IRCTC). Started with much fanfare in 2007 by the Ministry of Railways with support from the Ministry of Tourism, the train covers the eight destinations in a seven-day itinerary following the route shown in Figure 3.

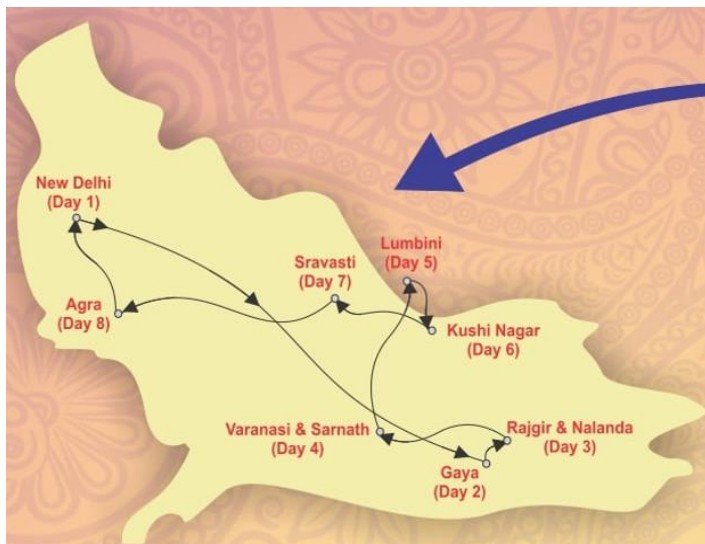

**Figure 3.** The itinerary of Mahaparinirvana Express (the Buddhist Circuit Train). Source: https://www.irctcbuddhisttrain.com/, accessed on 5 August 2020.

A concerted effort to promote Buddhist tourism began later with the discourse of circuits taking firm shape within the government policy. Before moving further, it is necessary to examine the circuits in more details with respect to visitation patterns.

## 6. Visitation Patterns in Buddhist Circuit

This section provides a nuanced understanding of visitation patterns in the main sites of the Buddhist circuit. These patterns are identified by using data from government reports and verified with the findings from the interviews.

The 2017 tourism datasets prepared by the Ministry of Tourism, India are used here as this is the only year that most comprehensive data is available for most sites and something that can be compared. The largest share of visitors is concentrated around the ten most iconic places [23]. Shown in Figure 4, these ten places account for close to 11 million visitors; almost 87% tourists are domestic whereas foreign visitors account for less than 13%. The number of foreigners varied across the key sites of the circuit: from as few as 26,900 in Sanksia to as high as 430,000 in Sarnath and 280,000 in Bodhgaya. The varying numbers show that not all foreign visitors visit all the eight destinations in the circuit. Visitation is uneven: Bodhgaya, Sarnath and Lumbini have visible presence of foreigners whereas other sites have much lesser numbers. In fact, some other sites outside the popular circuit such as Nalanda attracts sizable numbers of foreigners. Most international arrivals are predominantly from Buddhist dominated countries (Sri Lanka, Thailand, Japan, Korea, and China). This could be because these are countries with high Buddhist populations and have monasteries in Bodhgaya and Lumbini–which means that visitors have assured lodging and boarding.

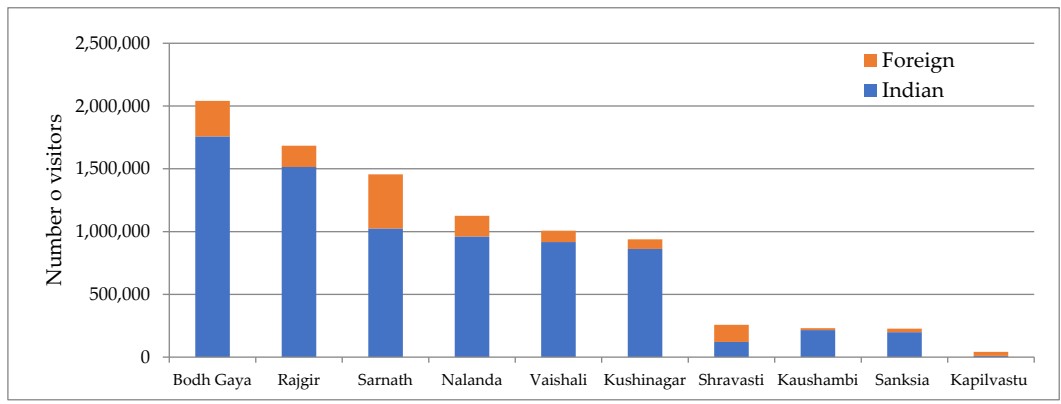

**Figure 4.** Number of visitors at key Buddhist sites, 2017. Source: Tourism statistics prepared by Ministry of Tourism, India [23].

A careful examination suggests a few salient characteristics of tourist flows. Except for Sarnath and Shravasti, foreigners constitute just between 8 percent to 15 percent of total visitor volumes in Buddhist sites while the rest is all non-Buddhist domestic tourists. It is the domestic tourists that really matter in terms of magnitude: about 1.75 million in Bodhgaya, about 1.5 million Nepalese in Lumbini. A visitor survey in Lumbini concluded that Lumbini is mainly "a destination for tourists" as "majority of visitors were domestic" and only "17% explicitly called themselves as pilgrims" Rai, 2018 [24]. In the Lumbini Sacred Garden, tourists engage in many kinds of hedonistic activities such as "boating in the lake, taking pictures of the exotic culture exhibits, posing in front of massive idols and eclectic statues of Buddha in the international monasteries. For many, the native architecture of monasteries itself is an exotic experience [24]" Bodhgaya does not have a designated sacred garden, but the monasteries zone has several monasteries that act more as a tourist attraction.

While questioning about the idea of circuit, all monks in their interviews highlighted that the circuitous journey to all the eight places is mainly undertaken by international Buddhist followers who are driven by the religious motive of pilgrimage where they would "undertake the necessary ritual practice, offer prayers, mediate, do prostrations and circumambulation" (Shinde). One monk summed it up: "foreigners are not here for leisure". But within foreigners also there is a distinction. A tour operator in Bodhgaya

pointed out that "maximum visitors are now from Thailand, Sri Lanka, Burma, and Vietnam but very little European because they are not interested in religious (dharmic) tourism." Such international pilgrims often travel in groups with organised tours. Another tour operator in Bodhgaya explained:

"The pilgrim-group would generally be associated with a monastery or Buddhist association, or a Buddhist school and that institution would organize their travel. The group would hire a commercial tour operator to organize transport but would bring their own guides or *bhante* (monks) from their home countries or hire locally. These would act as tour-leaders who will guide the itinerary and conduct of the group. Sometimes monks who stay at the international monasteries may assist."

Completing the circuit may take anywhere between 14 days to 29 days but many undertake only parts of it depending on several factors; some do right-hand circuit, and some do left-hand circuit (such as Sarnath-Bodhgaya-Sravasti) (tour operator, interviewed in Sarnath). There is not enough evidence to suggest clear ritual instructions about how to do the circuit even in the traditional scriptures which talk mainly about the sites rather than an order or sequence. These observations resonate what a report had identified in 2014: "the duration of stay in the Buddhist Circuit ranged from five to thirteen days (longer stays were recorded for tourists who arrived on packages) [5]." The cost of completing the circuit is quite high and most pilgrims travel on smaller budgets (hotel manager interviewed in Kushinagar). Those visiting Sarnath and Kushinagar are the ones who generally undertake the full circuit. Tour operators estimate that it is around 200,000 visitors per season that are on a package tour that can be called circuit-goers. Others spend more time in Bodhgaya and Lumbini that have better facilities including airports and direct flights, and most importantly more monasteries providing lodging and boarding. The president of hotel association in Bodhgaya remarked that "more than 80% visitors go to monasteries; less than 20% is at hotels". According to the monk at the Thai temple in Bodhgaya, international monasteries "make them feel-at-home with food, culture, language from their respective countries". He noted that "about 5000 visitors come to the wat in a year; live here in monasteries' for a maximum of 10 days; first time they come, they go to the circuit; after second visit they will come every year only to Bodhgaya." Many groups are affiliated with certain monasteries so "those attached with a religious leader will go to their facilities, why will they stay elsewhere" (hotel manager interviewed in Bodhgaya).

Thus, monasteries and private tour operators are more active in making the circuit work. It was found that Mahaparinirvan express-the much-hyped train circuit of the government "was not working" (interview with the Chairman of Hotel Association in Bodhgaya). In Kushinagar, none of the interviewees was even aware of the train and none in Sarnath had welcomed guests from this train.

From the interviews, it was clear that a large proportion of visitors (both pilgrims and tourists) do not complete the full circuit. Even within Buddhist followers, "Mahayaan are less in the circuit because they are not required to do the circuit...it is mostly the Theravadas because as per their teaching they should complete the circuit at least once . . . so it is the "very devoted" ones that go all the circuit" (owner of a Guesthouse in Bodhgaya). Thus, the actual numbers travelling the circuit is limited and that too concentrated around places with monasteries. For example, for many groups "Kushinagar would be a day-trip on the way to Lumbini". It was repeatedly pointed out that "domestic tourists do not travel the pilgrimage circuit" (tour operator interviewed in Bodhgaya).

According to a 2014 survey, the primary reason for Indian domestic visitors to visit Buddhist sites was "spiritual/religious travel" followed by "leisure and entertainment" and that the latter was cited as a primary reason by a large proportion of visitors (between 40 and 50 percent) at sites, such as Rajgir, Nalanda, and Sarnath [5]. This motivational difference may be mainly because these sites have sizable archaeological parks that possess numerous sites spread over large land tracts which can provide recreational relief and retreat for the day-trippers from nearby urban centres. In places such as Bodh Gaya and Lumbini, the plethora of international monasteries provide good resources for tourism [4,24].

Geographically, much of Buddhist tourism is concentrated around Bodhgaya, Rajgir, and Sarnath: this triangle collectively attracted more than 90 percent of Indian visitors and more than 83 percent of foreign visitors [5]. The IFC World Bank report further observed that tourists visit places around three main routes:

1. Bodhgaya-Rajgir-Nalanda-Kushinagar-Varanasi/Sarnath-Bodhgaya.
2. Bodhgaya-Rajgir-Nalanda-Sarnath-Kushinagar-Varanasi; and
3. Varanasi/Sarnath-Kushinagar-Bodhgaya-Varanasi.

These routes were also highlighted in the interviews with tour operators. One operator interviewed in Bodhgaya remarked that "these sites have become popular and are preferred because of their physical proximity and connections". All of them are within a range of 100 km (or two-hour drive) from Patna, the capital city of Bihar. Bodh Gaya is located about 105 km to the east; Rajgir is situated at about similar distance from Patna and 80 km from Bodh Gaya; Nalanda is 15 km away from Rajgir; Vaishali is at about 30 km from Patna towards Kushinagar; Kushinagar being the furthest at about 200 km towards west from Patna. A report in 2005 found that most tourists do not visit Buddhist destinations outside Bihar and awareness about sites outside of state is low: only 8% of tourists visiting Bihar had visited Sarnath which is in Uttar Pradesh [25]. Besides accessibility, the other reason for the concentration of tourist activity in these sites was the heavy presence of international monasteries (Bodhgaya has 150+; Lumbini has 44; Sarnath has about 12) that present a vibrant Buddhist culture and attract a continuous stream of international visitors.

Based on these findings it can be firmly established that large proportions of visitors are single site visitors rather than following the circuit. Except for the three sites–Bodhgaya, Sarnath, and Lumbini, other Buddhist sites are largely used as "middle-of-the-trip centers" or places for brief day visits by many visitors–international and domestic. Yet, the idea of circuit continues to feature strongly into the government discourse about Buddhist tourism.

## 7. The Government Discourse around Buddhist Circuits

This section examines the strategy of circuits as the main promotional tool for Buddhist tourism in India. Tourism is a subject in the Concurrent list meaning both central and state government have reasonable control and responsibility towards the sector and hence this discussion considers policies and projects from both tiers of governments.

With religious tourism as the main form of domestic tourism in India [12], it is natural that "religious circuits have emerged as an important framework for tourism development" in the country [9]. The significance of religious and cultural heritage was recognized in the first comprehensive tourism policy formulated in 1982 and tourism departments began to design "selective 'travel circuits' . . . to maximize the benefits of tourism" [26]. It must be noted that many Hindu pilgrimage circuits have been quite popular even without government's promotion [12,27]. However, Buddhist heritage and pilgrimage received much less importance owing to socio-cultural dynamics and politics of Buddhist as a minority group in India [6]. The increasing volume of international visitors and the lure of foreign dollars seem to be the primary motivators for the Government to focus on Buddhist sites for tourism development [5,9]. The other reason for Buddhist sites gaining government attention is to promote the diversity of cultural heritage and shared and syncretic relation between Hindu and Buddhist cultural heritage and addressing Buddhism as part of Indianness.

Earliest attempts to promote Buddhist circuit for tourism are seen in the "1986 -Action Plan for the Development of the Buddhist Sector" whose primary objective was to "identify exact locations at which accommodation and mid-way facilities are required to be put up" and where to create adequate travel infrastructure [28]. While the plan can be considered as a first, its implementation was slow, if any, and since then there have been only "fragmented efforts to develop and promote Buddhist Circuit as a mainstream tourism product" [5].

Given the increasing global interest in Buddhism, in early 2000s multi-lateral and intergovernmental agencies began to propose developing regional circuits across Asia. Under the South Asia Subregional Economic Cooperation (SASEC) program, Asian Devel-

opment Bank proposed three subthemes for development of regional Buddhist circuits in the subcontinent: Footsteps of Lord Buddha (comprising a cluster of sites including the eight key sites), Living Buddhism (developed around Buddhist places in northern parts of India for visitors to seek religious and spiritual experiences through yoga, meditation, and participation in festivals and retreats), and Art and Archaeology circuit [8]. According to the project report, "The integrated Buddhist Circuit concept has a brand image and visual identity that highlight its uniqueness, distinguishes it from other Circuits and destinations in India, and can be used to signpost the Circuit and build a sense of place" [8].

The regional cooperation provided an impetus to the 'Circuits' idea which was pursued vehemently by the Indian government (mainly through Buddhist Conclaves). Based on an agreement between the Union Ministry of Tourism, the Departments of Tourism of the Governments of Bihar and Uttar Pradesh, and the International Finance Corporation (World Bank Group, Bretton Woods, NH, USA), a major project called 'Investing in the Buddhist Circuit' was launched in 2013 with a vision "to transform 'a collection of sites' into a 'holistic tourism experience' that will help support revenue opportunities and jobs to improve quality of local life around the Buddhist sites" IFC, 2015 [9].

In 2015, the Ministry of Tourism, declared that it had "identified three Buddhist Circuits that will be developed with the help of Central Government/State Government/Private Stakeholders" [29]. The three circuits are described in Table 1 below. It is difficult to estimate how many days these circuits can take because these are only suggested itineraries and do not have any defined itineraries or visitor flows as yet.

**Table 1.** Circuits proposed by the Ministry of Tourism, Government of India. Source: Ministry of Tourism, [29].

| | | |
|---|---|---|
| Circuit 1 | Dharmayatra or Sacred Circuit | 5–7 days circuit including visits to Bodhgaya; Varanasi (Sarnath); Kushinagar; Kapilvastu; and a daytrip to Lumbini |
| Circuit 2 | Extended Dharmayatra or Sacred Circuit or Retracing Budhha's footsteps | 10–15 days circuit including visits to Bodhgaya (Nalanda, Rajgir, Barabar caves, Pragbodhi Hill, Gaya); Patna (Vaishali, Lauriya Nandangarh, Lauriya Areraj, Kesariya, Patna Museum); Varanasi (Sarnath); Kushinagar; Piparva (Kapilvastu, Shravasti, Sankisa); and a day trip to Lumbini |
| Circuit 3 | Buddhist heritage trails (State level circuits) | i. Jammu&Kashmir: Ladakh, Srinagar (Harwan, Parihaspora) and Jammu (Ambaran). ii. Himachal Pradesh: Dharamshala, Spiti, Kinnaur, Lahaul. iii. Punjab: Sanghon. iv. Haryana: Jind (Assam), Yamunanagar (Sugh). v. Maharashtra: Aurangabad (Ajanta, Ellora, Pithalkora Caves), Pune (Karla and Bhaja Caves), Mumbai (Kanheri Caves), Nashik (Pandavleni Caves). vi. Andhra Pradesh: Amravati, Nagarjunakonda, Vizag (Borra Caves, Salihundum Caves). vii. Madhya Pradesh: Sanchi, Satdhara, Andher, Sonari, Murulkurd. viii. Odisha: Dhauli, Ratnagiri, Lalitgiri, Udaygiri, Langudi, Khandagiri. ix. Chattisgarh: Sirpur. x. West Bengal: Kolkata (Indian Museum) xi. Sikkim: Rumtek, Enchay and other Monasteries. xii. Arunachal Pradesh: Tawang and Bomdila. |

The emphasis on these circuits should be contextualized with two other major initiatives. The first is the Swadesh Darshan launched in 2015 which focused on developing theme-based tourist circuits to cater to both mass and niche tourism. Five Pan-India Mega Circuits were identified including the Buddhist Mega Circuit that aims to showcase India as the Land of Buddha. Adding Buddha Circuit is more a matter of addressing the issues around cultural diversity within the country and to present Buddhist heritage in a better light to the neighboring countries that have higher Buddhist populations. Additionally,

thirteen theme-based tourist circuits were also identified of which the Buddhist Circuit, and the North-East India Circuit are relevant for the present study. The second initiative called PRASHAD (Pilgrimage Rejuvenation and Spiritual, Heritage Augmentation Drive) focused on the development, beautification, and augmenting of tourism infrastructure in pilgrimage places. The first phase focused on 13 cities of which three were related to Buddhist tourism (Varanasi, Gaya, and Amravati) [23]. Accordingly, several infrastructure projects for development of Buddhist sites in six state-level circuits (in Andhra Pradesh, Arunachal Pradesh, Bihar, Madhya Pradesh, Gujarat, and Uttar Pradesh) were initiated with substantial financial outlays under the Swadesh Darshan scheme [23]. Similarly, tourism authorities in many states are directly undertaking major improvement projects along circuits. For instance, in Odisha, the Odisha Tourism Development Corporation is renovating the existing accommodation facilities and adding new tourist facilities near the destinations of Lalitgiri, Ratnagiri, Udaigiri, Langudi, Dhauli and Jirang. However, there has been limited progress on most projects in both these schemes [30,31].

## 8. Challenges with Circuits as a Strategy

The popular Buddhist circuit in India is based on "a set of known, fixed sites associated with a singular tradition of Buddhism" that resulted from "archaeological rediscovery and restoration" [9]. This idea of Buddhist circuit (s) frames a certain religious and cultural imagination of the land of Buddha [10]. The Buddha's land is imagined as spread all over the Indian subcontinent because of the spread of Buddhism. However, the actual pilgrimage landscape of Buddha covers a specific geographic area and the sites discussed above. Although all sites in the circuit have their own sacred geographies, place-narratives, and symbols, these features are more enhanced in some sites than others depending on several factors including the scale of the place, demographics, tourism resources, and infrastructure. For instance, Bodhgaya is a major town and boasts of a long inventory of places related to Buddha. To this, many international monasteries add their interpretations for the clientele they serve augmenting the pilgrimage experience. However, in Kushinagar, except for the stupa of the reclining Buddha and the newly built Thai temple, visitors hardly go to any other site. Besides these two attractions, Kushinagar is largely a small settlement amidst of an agrarian landscape.

It could be argued that the circuit continues to anchor the pilgrimage landscape for the devout Buddhists. Additionally, such articulation provides the reference point for a well-accepted idea of a circuit into formal policy discourses and approaches to promotion and management of Buddhist heritage for tourism [7,28]. Geary's appeal of circuit as "key conduits for interconnection where the movement of people can be managed, serviced and coordinated through a concentration of sites and infrastructure" [9] is laudable and makes economic sense. However, as this study has found, for a large proportion of visitors this circuit is more like an imagined landscape made up of several destinations that they visit in any order or at any time.

From the above, it is clear that the act of travel is completed on certain parts of the recognised landscape and by a lesser proportion of visitors who recognise the circuit. While some religious monks claim to follow the traditional route and sequence, there is not enough evidence if there is indeed a correct sequence of visitation. In *Maha Parinibbana Sutta (Sutta N0.16)*, Buddha is shown as talking about four important places that correspond to the chronology of his life: birth, enlightenment, giving sermon, and death. While this may have some pattern during his lifetime, the geographical boundaries and access points may make it challenging to follow the same chronology. His birthplace is now in Nepal, the place of sermon is in the vicinity of the popular Hindu pilgrimage site of Banaras, and the place of enlightenment is in another state within India which has a bad repute for poor road access. Moreover, monks feel free to deviate from this order depending on the convenience of their patronising groups as far as they can cover as many places as possible in the given time. They claim that the modifications to the route and sequence does not affect the outcome of the pilgrimage as a religious practice or in connecting spiritually with Buddha

and Buddhist philosophy. The narration of place stories and meditative prayers are enough. So, from a ritual and religious sense, the circuits are okay. However, such an orientation is not seen in offerings of circuit tours by other stakeholders who are mainly focused on value for money by compressing as many destinations in shorter itineraries. By proposing circuits at the pan-India level, and then at the individual level, the government seem to try and establish the idea of a Buddhist landscape that covers the breadth of the country. However, as shown in this paper, these are not practical on several accounts including the distances between the sites within the circuit, the level of infrastructure development, state of archaeological resources, interpretation facilities, and the appeal of the sites for tourism in competition with other religious circuits.

Practically, and from a policy perspective, promotion of circuits as a tourism strategy can be challenged at least on two accounts.

### 8.1. Circuits for Foreigners but Destinations for Domestic Tourists

Both quantitative and qualitative differences are enormous in the way of foreign and domestic tourists in Buddhist sites. Although the global Buddhist population is around 450 to 480 million, but only 0.005 percent is visiting Buddhist sites in India IFC, 2015 [9]. In India, the volume of tourists at Buddhist sites is very limited: around 1.5 million foreign visitors and 11 million domestic tourists compared with an estimated 15.5 million foreign visitors and 1652.49 million domestic tourists in 2017 [23]. Even at Buddhist sites, the proportion of international visitors is small in volume compared to the numbers of domestic visitors. Beyond the circuit in Bihar, there are only two Buddhist places with significant tourist flows: the Amravati Stupa and Nagarjunakonda in Andhra Pradesh (south India) and Dharmshala, the residence of Dalai Lama in Himachal Pradesh recording 3.1 million and 1.5 million domestic tourists, respectively. Compare these with foreign visitors: only 1053 in Amravati, and 41,188 in Dharmshala. In other Buddhist sites, the numbers of tourists run only in thousands and hundreds which is very low compared to other religious and cultural tourism destinations in India [12].

Since the travel in the circuit is primarily motivated by religious practice and spiritual seeking, the magnitude of such international pilgrims in the circuit is even smaller. They cover the four main sites, but their itineraries are customized as per convenience of travel logistics offered by the organizers and therefore the inclusions of other sites varies considerably. Given the nature of Buddhist philosophy, even in those circuits, there is not necessarily a fixed order of visitation or ritual performance that are obligatory. It is speculated that in the circuit "the most lucrative spenders are international travelers, especially those coming from Western countries and China" [9] and hence the circuit should be yielding high returns. But as field data shows a large proportion of circuit travelers are Buddhist pilgrims who are relatively low spenders–something that was recognized in a high-level report as well [5]. Wherever religious circuits have become successful, it is because of the large numbers of followers of that religion [16,27]. That, however, is not the case with respect to Buddhism where the flow of domestic tourists is limited.

The idea of inventing more circuits, particularly, at state level, seems far-fetched. For instance, states such as Maharashtra are promoting sites as part of a larger circuit. This may sound plausible because at present of all Buddhist followers in the country, 77% are in Maharashtra. However, most of these follow the form of Buddhism that was embraced by national leader Babasaheb Ambedkar in 1950s [6]. Their religious practice revolves around places dedicated to Ambedkar including the Stupa at Nagpur and in Mumbai. All other sites identified by the government are archaeological sites and mostly in form of caves. If these are connected by a route it will be about a 650 kms long which is an unreasonable distance to travel for a circuit that does not offer a religious experience. While individually these sites have some visitation, there is no information available if anyone visits all these as a part of circuit or even a route. The idea of circuits being successful in states with even less Buddhist population seems even bleak–who and what can enliven the archaeological nature of those sites? Following existing trends, such sites may benefit from

being developed as destinations for one-day outings but with more sensitivity towards heritage for domestic tourists.

### 8.2. Embeddedness of Buddhist Sites in a Different Socio-Spatial Context

Most Buddhist sites are embedded in a socio-cultural milieu that includes multiple religious faiths such as Hinduism, Islam, and Christianity and tribal traditions in mountainous regions. This makes for complicated overlaps in which the Buddhist sites must survive and flourish. On one hand, the international Buddhist followership is often interested in maintaining "an imagined past of the place" [32]. On the other local communities are interested in the economy of Buddhist tourism and tourism-led-development. In a study of Bodhgaya, Geary makes a pointed observation, "the desire to fulfil the expectations of visitors seeking inward reflection, meditation and devotion, often comes at the expense of the urban poor, not to mention the remarkable inflation of land generated by major hotel chains and an expanding number of high financed Buddhist monasteries and temples" [9]. Ideological dilemmas such as development that favors some external parties at the cost of hotel industry, or lesser role for the engagement of local community against the preference for foreign stakeholders wanting to invest, there are also real practical challenges. In places such as Bodhgaya, preference is given to "new constructions by Buddhists from Tibet, Vietnam, Thailand, and other places in Asia within the new Master Plan zones" but local businesses face the wrath of being informal [32]. Suffice to note that for a more holistic outcome of Buddhist tourism, it is necessary to move away from the enclave tourism to a wider participation of local communities.

Many Buddhist sites are under the control of Archaeological Survey of India (ASI)-the apex central authority that is also tasked with the management of heritage structures. ASI has created archaeological park in these places and have stringent rules and regulations for the conservation of such sites by not allowing any active religious practice. This is observed in Sarnath and Kushinagar where pilgrims cannot perform any rituals at the stupas which disappoints Buddhist followers wanting to pray and pay homage to Buddha. In Sarnath, a monk shared his pain, "we cannot do any puja for our Buddha . . . and this leads to some incompleteness". For tourists also, there is limited opportunity of interpretations due to lack of guides. Buddhist archaeological sites, despite their tourism potential, are underutilized and wanting more visitors.

As a concept, circuits are good for doing tourism [9,18]. Yes, there is a merit in the circuit because that allows the seekers to fulfil their spiritual experience in following the footsteps of Buddha, but they need the guidance and mediation of monks and monasteries. For a large part, Buddhist circuits seem more like a route that supports destination-oriented travel rather than a circuitous and more fulfilling journey for mass tourists. Considering the existing travel patterns and the unique socio-spatial and cultural context in Buddhist sites, it seems prudent to revisit the emphasis on pilgrimage circuits as a strategy to promote and develop Buddhist tourism. A Thai monk offered advice: "What is a real need? For instance, making toilet is useless maintenance; major issue is road connection; need information centre; resolve border issues and problems; need to address concerns of religious leaders who are the ones to put life in these places". It is hard to sell the entire circuit as a tourism product because of the various challenges and therefore it would rather help to focus on Buddhist sites as destination and improve tourism infrastructure to make them more visitor friendly.

## 9. Conclusions

This paper began by asking questions about the articulation of Buddhist pilgrimage circuits as religious tourism product and their effectiveness as a strategy for promoting Buddhist heritage tourism in India. These questions are very relevant because India is a large repository of archaeological and living heritage of Buddhism and therefore offers several opportunities for an "authentic" Buddhist tourism experience [5,8,33]. In answering the key questions, this paper has shown that much of tourism for Buddhist heritage is

framed through Buddhist pilgrimage circuits because the idea of this circuit is rooted in the sojourns of Buddha and is embraced by religious institutions serving Buddhist teachings and values to their followers. However, this paper argues that Buddhist circuits represent a different value proposition (enshrined in Buddhist teachings) for ardent Buddhists and foreign tourists. But foreign visitation accounts for less than 15% of visitor flows to Buddhist sites, the remaining majority being domestic tourists. Having analyzed visitation patterns to Buddhist sites, the paper further contends that the idea of circuit becomes less meaningful for mass tourism as most domestic visitors (overnight and same day) undertake one-site visits; make their own travel arrangements themselves, and a very small percentage avail of package tours. This means that the idea of forced circuits may not be that effective. And therefore, articulation of circuits for domestic tourism should be seen with caution.

Following the national lead, most states in India seem to be pushing the agenda of circuits connecting sites within their jurisdictions for instance refer to [34–37]. As illustrated here, circuits as a strategy for promotion of Buddhist heritage for tourism is overemphasized and needs to be revisited and it is important to ask who the target audience for circuits is. It would be advisable to modify these circuits as "routes and destinations" for prioritizing decisions on investing in tourism infrastructure. For instance, developing interpretation facilities at key destinations should be a priority in archaeological sites and not "a *light and sound show* because of the spiritual nature of these sites–they reflect deeper knowledge and meaning" (monk interviewed in Sarnath). In principle, circuits and individual sites are expected to be places of spiritual experience for the visitors interested in Buddhism. Those undertaking a journey on the circuit are mainly motivated by spiritual and religious reasons. This volume, however, is negligible compared to the large numbers of domestic visitors and day-trippers to individual sites where the international monasteries and open landscapes of archaeological parks function as tourism attractions. It could be said that their leisure orientation may lead to touristy shallowness. Offering a "*light and sound show*" may not add any more value to both types of visitors in the site. Pilgrims are there for spiritual experience that is best gained through solitude and meditative practices with their gurus. A Buddhism-based *light and sound show* may add only a little recreational value, if at all to tourists. Moreover, a logistical challenge is that the light and sound show will have to be performed in the night. Buddhist sites have a limited hotel industry as most visitors who stay overnight, prefer accommodation in monasteries. Thus, in concept and practice, a *light and sound show* as a strategy to promote Buddhist tourism needs more work.

While circuits may serve for image-building, consistent branding, and packaging of Buddhist sites for economic benefit of the state, they should follow a realistic and comprehensive approach so that the real benefits of circuitous development can be realized for sustainable development of Buddhist tourism.

The current discourse of circuits does not adequately acknowledge the active role of international associations and societies of Buddhist followers in Buddhist sites (namely Bodhgaya, Rajgir, Sarnath and so on) in maintaining the regular flows of international visitors. The significance of such international organizations also means that there is a kind of enclave tourism where the visitors tend to be removed from the local context. This means that visitors are using lodging and boarding facilities in monasteries and patronizing them with donations rather than contributing to local economy in the town–the benefits are unequal. Often in such situations, conflicts are latent and need to be addressed for tourism to sustain [4]. There is also a potential for contestation in religious sense as "the government looks to regulate and reproduce a sacred geography" which means there will be "tensions surrounding the ritual activities associated with Buddhist pilgrimage" [9]. The problems presented by the monks regarding their inability to perform their religious practices in Sarnath and Kushinagar are illustrative and are applicable to most Buddhist sites which face similar situation of being archeological sites controlled by ASI. It is important to make better use of Buddhist archaeological sites in a way that can help in their restoration and conservation. The active and living sites of Buddhist practices situated in smaller fragile

ecological and cultural regions need to be safeguarded against the negative impacts of increasing tourism that may soon be beyond their carrying capacities [5,8,22,38].

More broadly, the paper questions the translation of imagined cultural landscapes into physical circuits for tourism. There is a difference between the imagined circuit and the physical circuits; the former is more related to cognition of a sacred landscape while the latter is based on access and convenience of logistics of transport and accommodation. The practicalities of travelling to the sites supersede the need to cover all the sites enroute of the circuit. Religious merit is attributed to visiting and investing more time in the sites with religious gurus and monks rather than adherence to sequencing (which anyway is not sanctioned as a religious practice) in terms of time and route and therefore the circuit becomes more of a collection of sites connected with a flexible itinerary. So, circuits are okay in religious and practical sense, but the bigger concern is the differential tourist flows that prefer destinations as to the complete circuit. And hence it is necessary to focus on destinations simultaneous to the idea of a Buddhist circuit for promoting Buddhist heritage. The explanations of cultural landscapes are about encountering the divine and the sacred through the journey, but the economic rationale of a circuit lies in the management of the infrastructure necessary for completing the travel. While the cognitive imagination anchors the sacred geography, to make the circuit work as a tourism product offering a holistic experience, considerable effort is required in presenting all sites with similar rigor and similar level of visitor facilities.

In challenging the notions of Buddhist circuits, this paper has only sparingly looked at many issues that may have a significant bearing on a comprehensive understanding of Buddhist heritage. A more nuanced work on tour operators offering Buddhist packages will help in understanding the organizing of circuit tourism better. Similarly, focused studies of religious institutions such as monks and monasteries have the potential to illuminate the religious and spiritual dimensions of tourism to Buddhist sites [12,21,22]. As more and more organizations offer well-being-oriented tourism around meditation, yoga, Buddhist teaching and philosophies in Buddhist places there is much more to Buddhist heritage than circuits.

**Funding:** This research received no external funding. The fieldwork for this research was funded by La Trobe University's Internal Grant.

**Institutional Review Board Statement:** The study was conducted in accordance with the Declaration of Helsinki and approved by the Institutional Review Board (or Ethics Committee) of La Trobe University, Australia (HEC19393 dated 15/10/2019) for studies involving humans.

**Informed Consent Statement:** Informed consent was obtained from all subjects involved in the study.

**Data Availability Statement:** Not applicable.

**Conflicts of Interest:** The authors declare no conflict of interest. The funders had no role in the design of the study; in the collection, analyses, or interpretation of data; in the writing of the manuscript; or in the decision to publish the results.

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
