# Peer review of "How Effective Is a Buddhist Pilgrimage Circuit as a Product and Strategy for Heritage Tourism in India?"

_heritage, doi:10.3390/heritage5040198_

Round 1
Reviewer 1 Report
The article focuses on the Buddhist heritage, namely Buddhist circuits as a tourism product. The topic of the article is interesting but the article has some flaws, primarily in the presentation. Suggestions are put forward below.
The author should dedicate a special section to Methodology. It is mentioned, but not sufficiently explained. It is necessary to describe the survey (what was its goal? what type of questions it consisted of?) as well as the participants: out of 75 interviewees, how many were monks, tour operators, and government employees? How was the survey taken (in person? digitally?).
Along the same line, there is no clear Results section. It seems that section 5. actually acts as Results section. However, results are not clearly presented. Rather, they seem like a series of direct quotations from the interviewees but without clear outline. If the author manages to clearly explain the Methodology, then Results should easily follow. It is suggested to put some more work in it. Only then it should be put in the relation with the existing documentation as well as theory in the Discussion section.
Figure 1. - if possible, enhance it, it is not clearly visible.
299-302 - should be deleted; I guess it is the text pertaining to the Word template which remained.
Some language editing is required (e.g. " This paper aims to examines"; 225 - "but" is not necessary; 274 - space needed between or and 10; etc.)
Author Response
Thank you for your valuable comments. I have responded to the comments as track changes in the manuscript and summarised them here.
|
# |
Reviewer comment |
Author’s response |
|
1 |
The author should dedicate a special section to Methodology. It is mentioned, but not sufficiently explained. It is necessary to describe the survey (what was its goal? what type of questions it consisted of?) as well as the participants: out of 75 interviewees, how many were monks, tour operators, and government employees? How was the survey taken (in person? digitally?) |
Thank you for this suggestion. A succinct section on methodology is added:
4. Methodology The fieldwork was conducted in the main sites of the Buddhist circuit – Bodhgaya, Lumbini, Sarnath, and Kushinagar in December 2019. Besides participant observation, interviews with key stakeholders were undertaken to understand the visitation patterns, the rituals and performances undertaken by pilgrims, and overall, the pilgrimage economy and how it impacted the local community. Using a snowball method (the author contacted the managers of temple/ monastery trusts in these places and with their reference recruited potential interviewees), a total of 75 interviews were con-ducted covering representatives of the main stakeholders including 26 monks, 21 hotel managers, 12 government officials, 10 local community members, and 6 tour organisers. Each stakeholder was given a customized interview guide for interview but some of the common questions asked were: What are different elements of Buddhist heritage and Buddhist pilgrimage in this place; where do most visitors come from and what are their activities; what changes have they observed in Buddhist pilgrimages over time; and so on to get a holistic understanding of Buddhist heritage and pilgrimage. The 45-minutes interviews were conducted in person at the place of work of the stakeholders. |
|
|
Along the same line, there is no clear Results section. It seems that section 5. actually acts as Results section. However, results are not clearly presented. Rather, they seem like a series of direct quotations from the interviewees but without clear outline. If the author manages to clearly explain the Methodology, then Results should easily follow. It is suggested to put some more work in it. Only then it should be put in the relation with the existing documentation as well as theory in the Discussion section. |
Thank you for the comment. Now that a separate section (section 4) is created for Methodology, it is clear that the following sections (5,6,7) are all indicating the results. This has been made even explicit by changing the opening sentence in Section 5 to:
“Based on the findings from the fieldwork, at least three stakeholders are identified as presenting and promoting the Buddhist pilgrimage circuit: religious actors (monks and monasteries), private tour operators, and government agencies.”
I believe the content about the circuit goes beyond “existing documentation” and present analytical insight as no other work has made those connections between how stakeholders present their version of circuits.
In section 6, the opening sentence is changed to relate findings with methodology.
“This section provides a nuanced understanding of visitation patterns in the main sites of the Buddhist circuit. These patterns are identified by using data from government reports and verified with the findings from the interviews.”
Overall, I believe that the presentation of key findings in the existing narrative format is much better suited to the purpose of the paper- this is something that other reviewers have also appreciated. |
|
|
Figure 1. - if possible, enhance it, it is not clearly visible |
This is the best resolution possible so cannot enhance it. |
|
|
299-302 - should be deleted; I guess it is the text pertaining to the Word template which remained. |
That was an error. Deleted. |
|
|
Some language editing is required (e.g. " This paper aims to examines"; 225 - "but" is not necessary; 274 - space needed between or and 10; etc.) |
All these minor errors corrected. |
Reviewer 2 Report
This is a well written article providing plenty of information about the reality of Buddhist tourism and circuits and how the actual picture diverges from official strategies seeking further development for economic purposes. The language is very good and fluent (few words are missing, see red highlights), subparts of the article are well articulated with each other. There is a good use of references to illustrate the ideas, including many websites and an adequate explanation of used concepts.
However,
I missed more analytical thinking that would connect the facts and figures with religious concepts by for instance digging further into the circumambulation question, or the discrepancy between circuits as religious experience vs. economic motivations, etc, see my comments in the text. You wrote in the conclusion: “the paper questions the translation of imagined cultural landscapes into physical circuits for tourism”, but you haven’t really “questioned” this translation, just given the facts without really analyzing them = is there a difference between the imagined and the physical circuits here? What is different and why? Is it more or less valuable in religious terms if the physical is different from the imagined? Maybe also emphasize the clash between different views, from the religious authorities on the one hand, and the gvt and tourist agencies on the other; and what do the Buddhist followers say about that since they are at the junction between these two views, they have to use the available facilities that lead them to specific places that are touristically promoted, but what about the other places with no facilities and promotion? Do the followers feel they miss something at the religious level if they don’t travel to those other places? I understood that circuits are fixed for questions of practicality and money, but are these circuits ok in ritual and religious terms?? That would also be interesting among other things!!
I therefore recommend publication of this article provided it receives a bit more analysis to make it thicker from the research point of view. Specific places have been highlighted in yellow in the text.
All the best!

Author Response
Thank you for your valuable suggestions. I have responded to all comments as track changes in the manuscript as well besides the responses here.
|
# |
Reviewer Comment |
Author’s response |
|
1 |
This is a well written article providing plenty of information about the reality of Buddhist tourism and circuits and how the actual picture diverges from official strategies seeking further development for economic purposes. The language is very good and fluent (few words are missing, see red highlights), subparts of the article are well articulated with each other. There is a good use of references to illustrate the ideas, including many websites and an adequate explanation of used concepts. |
Thank you for an overall appreciating and encouraging comment. All the minor errors are corrected. |
|
2 |
I missed more analytical thinking that would connect the facts and figures with religious concepts by for instance digging further into the circumambulation question, or the discrepancy between circuits as religious experience vs. economic motivations, etc, see my comments in the text. |
Agreed. Some rewriting has been done to reflect the analysis. All the responses to the comments are tracked in the manuscript. The following sentences have been added as para 3 in the discussion section (section 8) to make the analytical relationships more explicit.
From the above, it is clear that the act of travel is completed on certain parts of the recognised landscape and by a lesser proportion of people who recognise the circuit. While some religious monks claim to follow the traditional route and sequence, there is not enough evidence if there is indeed a correct sequence of visitation. In Maha Parinibbana Sutta (Sutta N0.16), Buddha is shown as talking about four important places that correspond to the chronology of his life: birth, enlightenment, giving sermon, and death. While this may have some pattern during his lifetime, the geographical boundaries and access points may make it challenging to follow the same chronology. His birthplace is now in Nepal, the place of sermon is in the vicinity of the popular Hindu pilgrimage site of Banaras, and the place of enlightenment is in another state within India which has a bad repute for poor road access. Moreover, monks feel free to deviate from this order depending on the convenience of their patronising groups as far as they can cover as many places as possible in the given time. They claim that the modifications to the route and sequence does not affect the outcome of the pilgrimage as a religious practice or in connecting spiritually with Buddha and Buddhist philosophy. The narration of place stories and meditative prayers are enough. So, from a ritual and religious sense, the circuits are okay. However, such an orientation is not seen in offerings of circuit tours by other stakeholders who are mainly focused on value for money by compressing as many destinations in shorter itineraries. By proposing circuits at the pan-India level, and then at the individual level, the government seem to try and establish the idea of a Buddhist landscape that covers the breadth of the country. But as shown in this paper, these are not practical on several accounts including the distances between the sites within the circuit, the level of infrastructure development, state of archaeological resources, interpretation facilities, and the appeal of the sites for tourism in competition with other religious circuits. |
|
3 |
You wrote in the conclusion: “the paper questions the translation of imagined cultural landscapes into physical circuits for tourism”, but you haven’t really “questioned” this translation, just given the facts without really analyzing them = is there a difference between the imagined and the physical circuits here? What is different and why? Is it more or less valuable in religious terms if the physical is different from the imagined? Maybe also emphasize the clash between different views, from the religious authorities on the one hand, and the gvt and tourist agencies on the other; and what do the Buddhist followers say about that since they are at the junction between these two views, they have to use the available facilities that lead them to specific places that are touristically promoted, but what about the other places with no facilities and promotion? Do the followers feel they miss something at the religious level if they don’t travel to those other places? I understood that circuits are fixed for questions of practicality and money, but are these circuits ok in ritual and religious terms?? That would also be interesting among other things!! |
Thank you for the suggestion. As mentioned in the previous point, more analytical statements have been added in the discussion section. They support this conclusion about translation. Once again, following sentences are added to reinforce the explicit connection.
There is a difference between the imagined circuit and the physical circuits; the former is more related to cognition of a sacred landscape while the latter is based on access and convenience of logistics of transport and accommodation. The practicalities of travelling to the sites supersede the need to cover all the sites enroute of the circuit. Religious merit is attributed to visiting and investing more time in the sites with religious gurus and monks rather than adherence to sequencing (which anyway is not sanctioned as a religious practice) in terms of time and route and therefore the circuit becomes more of a collection of sites connected with a flexible itinerary. So, circuits are okay in religious and practical sense, but the bigger concern is the differential tourist flows that prefer destinations as to the complete circuit. And hence it is necessary to focus on destinations simultaneous to the idea of a Buddhist circuit for promoting Buddhist heritage.
|
Reviewer 3 Report
The article is a pleasure to read, its critical conclusions are well-founded and illustrated.
The contrasting of the idea of the 'circuit' with data and interviews focussed on the lived experience of tourists and heritage professionals works really well. Excellent work and I wholeheartedly recommend publishing this article with (very) minor revisions.
One question that came up whilst reading is that the article's suggestions might be a bit speculative. For instance, in lines 628, a 'light and sound show' is criticized as a way to further develop the circuit. Instead, the author follows the interviewed monk in suggesting that something more reflective of the spiritual nature of these sites might be more in order. This is mirrored in the conclusions, for instance in lines 604/605 an alternative to the circuit is suggested: 'it would rather help to focus on Buddhist sites as destination' (instead of as elements of a circuit'.
Now, I am fully on board with the criticism that a Circuit brings along infrastructural issues that clash with the logistic preferences of the majority of those visiting the sites. But what is the link between organizing Circuits and the 'light and sound show'? Might it, at least in theory, not also be possible to have a circuit focus on the religious/spiritual nature of these sites?
It might still be that a circuit clashes with the logistic preferences of the actual visitors (who might visit only one day, for instance), but I don't think we know about what the tourists in question think about the light and sound shows right? Or, to parafrase my question, presenting sites as individual destinations might not mean that the more mundane touristy elements of tourism are kept at bay right? It seems that there is a connection set up in this article between touristy-shallowness and the Circuit on the one hand, and a more profound spiritual tourism oriented towards the individual sites on the other.
Although broken down in its parts, the conclusions are fully convincing, the equation of 'Circuit' with shallow tourism and orientation towards spiritually more profound presentation of destination with the emphasis on individual sites, seems to need a bit more work to be fully coherent (do we know whether the visitors reject the 'touristy' elements of the circuit? Maybe they would like a light and sound show AND a presentation oriented towards individual sites?
But these are minor comments in what remains a very fine paper indeed
Author Response
Thank you for your valuable and encouraging comments. I have responded to those as track changes in the revised manuscript and summarised here.
|
# |
Reviewer Comment |
Author’s response |
|
1 |
The article is a pleasure to read, its critical conclusions are well-founded and illustrated. The contrasting of the idea of the 'circuit' with data and interviews focussed on the lived experience of tourists and heritage professionals works really well. Excellent work and I wholeheartedly recommend publishing this article with (very) minor revisions. |
Thank you for the appreciative and encouraging comments. |
|
2 |
One question that came up whilst reading is that the article's suggestions might be a bit speculative. For instance, in lines 628, a 'light and sound show' is criticized as a way to further develop the circuit. Instead, the author follows the interviewed monk in suggesting that something more reflective of the spiritual nature of these sites might be more in order. This is mirrored in the conclusions, for instance in lines 604/605 an alternative to the circuit is suggested: 'it would rather help to focus on Buddhist sites as destination' (instead of as elements of a circuit'. Although broken down in its parts, the conclusions are fully convincing, the equation of 'Circuit' with shallow tourism and orientation towards spiritually more profound presentation of destination with the emphasis on individual sites, seems to need a bit more work to be fully coherent (do we know whether the visitors reject the 'touristy' elements of the circuit? Maybe they would like a light and sound show AND a presentation oriented towards individual sites? |
I am not sure if I understand this comment about the link between organizing circuits and the “light and sound show”. Nonetheless, I have added the following sentences to provide more analysis and clarification about the futility of light and sound show.
In principle, circuits and individual sites are expected to be places of spiritual experience for the visitors interested in Buddhism. Those undertaking a journey on the circuit are mainly motivated by spiritual and religious reasons. This volume, however, is negligible compared to the large numbers of domestic visitors and day-trippers to individual sites where the international monasteries and open landscapes of archaeological parks function as tourism attractions. It could be said that their leisure orientation may lead to touristy shallowness. Offering a “light and sound show” may not add any more value to both types of visitors in the site. Pilgrims are there for spiritual experience that is best gained through solitude and meditative practices with their gurus. A Buddhism-based light and sound show may add only a little recreational value, if at all to tourists. Moreover, a logistical challenge is that the light and sound show will have to be performed in the night. Buddhist sites have a limited hotel industry as most visitors who stay overnight, prefer accommodation in monasteries. Thus, in concept and practice, a light and sound show as a strategy to promote Buddhist tourism needs more work. |
|
|
But these are minor comments in what remains a very fine paper indeed |
Thank you for this comment. Much appreciated. |
Round 2
Reviewer 1 Report
The authors have not addressed all the suggestions: the research questions, assumptions or hypotheses are still missing. Also, although the methodology section was improved, the authors still did not address the suggestion of stating a goal of the survey. Finally, the research questions should be replied to. Regarding Figure 1, the suggestion is to make a new map since the one presented is of extremely low quality and adds nothing to the research. The other figures are also of a low quality, but this one in particular.
Author Response
Dear reviewer,
Thank you for the comments to improve my paper. I have responded to those comments in revising the manuscript. Here is an overview of those responses.
|
# |
Reviewer comment |
Author response |
|
1 |
the research questions, assumptions or hypotheses are still missing |
I have added the following statement about research question in para 4 of Introduction.
This paper has two research questions: a) do traditional Buddhist pilgrimage circuits make for a good religious tourism product, and b) whether pilgrimage circuits are an effective strategy to promote Buddhist heritage for tourism in India. To find answers to these questions, the paper first examines the articulation of the main Buddhist pilgrimage circuit in terms of places, place-stories, practices, and itineraries based on religious traditions. It uses the conceptual approach of cultural landscape and the geographical underpinnings of a circuit to explain the articulation of Buddhist pilgrimage circuits. Then, it investigates the visitation patterns in the different sites that constitute the pilgrimage circuit to understand if the pilgrimage circuit really works as a circuit. While doing so, the paper also challenges the assumptions about incorporating the idea of the traditional pilgrimage circuit as a main strategy into the tourism policy. By answering these questions, the paper offers a nuanced understanding of Buddhist pilgrimage circuits and their relevance for tourism in India – both as a concept, and as a practical strategy, for tourism development in Buddhist sites.
(In this study, I really do not think that an explicit hypothesis is required as I am discussing ideas and assumptions about traditional pilgrimage circuits and challenging those based on evidence from fieldwork about contemporary patterns of travel. It is common in qualitative studies to use approaches that are different from the methods of approving or disapproving a hypothesis in quantitative research. I do not feel the need to state an explicit hypothesis and trying to prove or disapprove it as this is something beyond the focus and purpose of this paper. The presentation of research data and its analysis through a narrative style is the strength of this paper. And this is something that is clearly appreciated by other reviewers as well. Using the approach of a hypothesis may lead to writing of another paper.) |
|
2 |
although the methodology section was improved, the authors still did not address the suggestion of stating a goal of the survey |
I think the comment about surveys would be relevant if my study was using a quantitative approach – which it is not. I would like to restate that I did not conduct any surveys. I conducted detailed in-depth interviews for more holistic understanding of how the pilgrimage circuit works. To clarify this further, I have added the following sentence in the methodology section.
The fieldwork included participant observation and interviews with key stakeholders. The goal of the interviews was to understand the working of the pilgrimage circuit and how it intersected with the tourism in its constituent sites. As such the questions were asked around different elements of Buddhist heritage and Buddhist pilgrimage in those sites, patterns of visitation, rituals and performances undertaken by pilgrims, activities of tourists, roles of different stakeholders in the pilgrimage economy, the impacts of tourists on the local community, and the changes observed in Buddhist pilgrimages and pilgrimage sites over time. |
|
3 |
the research questions should be replied to |
I think the discussion and conclusion sections have answered the main research question (although in a narrative style). I have signposted this again with an addition and rephrasing of the following paragraph in the conclusion section.
This paper began by asking questions about the articulation of Buddhist pilgrimage circuits as religious tourism product and their effectiveness as a strategy for promoting Buddhist heritage tourism in India. These questions are very relevant because India is a large repository of archaeological and living heritage of Buddhism and therefore offers several opportunities for an “authentic” Buddhist tourism experience [5,8,33]. In answering the key questions, this paper has shown that much of tourism for Buddhist heritage is framed through Buddhist pilgrimage circuits because the idea of this circuit is rooted in the sojourns of Buddha and is embraced by religious institutions serving Buddhist teachings and values to their followers. However, this paper argues that Buddhist circuits represent a different value proposition (enshrined in Buddhist teachings) for ardent Buddhists and foreign tourists. But foreign visitation accounts for less than 15% of visitor flows to Buddhist sites, the remaining majority being domestic tourists. Having analyzed visitation patterns to Buddhist sites, the paper further contends that the idea of circuit becomes less meaningful for mass tourism as most domestic visitors (overnight and same day) undertake one-site visits; make their own travel arrangements themselves, and a very small percentage avail of package tours. This means that the idea of forced circuits may not be that effective. And therefore, articulation of circuits for domestic tourism should be seen with caution.
(I sincerely believe this rephrasing is reasonable in addressing the reviewer’s comment without losing the essence and arguments of the paper (that have been applauded and appreciated by other reviewers too). |
|
4 |
the suggestion is to make a new map since the one presented is of extremely low quality and adds nothing to the research |
I agree with these comments. Accordingly, I have removed Figure 1 and rephrased the sentence in section 5.1 as:
According to this itinerary, the unidirectional tour starts at Sarnath which is considered as the birthplace of the sublime dhamma and follows through to Bodhgaya; Gijjakuta (Vultures' Peak); Rājagaha (Rajgir); Nalanda; Vaishāli (Vishālā); Kusinārā (Kushinagar); Lumbini, Kapilavastu, (in Nepal); Sāvatthi (Jetavanārāma) (or Sravasti); and ends in Sankassa and then pilgrims go out of the country from Delhi. |